# How Reliable Is Fluorescence-Guided Surgery in Low-Grade Gliomas? A Systematic Review Concerning Different Fluorophores

**DOI:** 10.3390/cancers15164130

**Published:** 2023-08-16

**Authors:** Andrea Bianconi, Marta Bonada, Pietro Zeppa, Stefano Colonna, Fulvio Tartara, Antonio Melcarne, Diego Garbossa, Fabio Cofano

**Affiliations:** 1Neurosurgery, Department of Neurosciences, University of Turin, 10126 Turin, Italy; marta.bonada@edu.unito.it (M.B.); pietro_zeppa@yahoo.it (P.Z.); amelcarne@cittadellasalute.to.it (A.M.); diego.garbossa@unito.it (D.G.); fabio.cofano@unito.it (F.C.); 2Headache Science and Neurorehabilitation Center, IRCCS Mondino Foundation, Department of Brain and Behavioral Sciences, University of Pavia, 27100 Pavia, Italy; 3Humanitas Gradenigo, 10100 Turin, Italy

**Keywords:** low-grade glioma, fluorescence, sodium fluorescein, 5-aminolevulinic acid, glioma, intraoperative fluorescence

## Abstract

**Simple Summary:**

Low-grade gliomas account for approximately 10–20% of all primary brain tumors. Many techniques have been proposed to achieve a maximal safe resection, and among them is fluorescence-guided surgery. Currently, the most widely used fluorescent-labeling dye is 5-aminolevulinic acid, particularly used for high-grade gliomas, while other agents have been proposed but not extensively assessed. Therefore, the aim of our systematic review is to synthesize the actual evidence in order to highlight whether fluorescence-guided resection is reliable even in low-grade gliomas and, among the various fluorophores, which one is the best in highlighting tumor boundaries.

**Abstract:**

Background: Fluorescence-guided surgery has been increasingly used to support glioma surgery with the purpose of obtaining a maximal safe resection, in particular in high-grade gliomas, while its role is less definitely assessed in low-grade gliomas. Methods: A systematic review was conducted. 5-aminolevulinic acid, sodium fluorescein, indocyanine green and tozuleristide were taken into account. The main considered outcome was the fluorescence rate, defined as the number of patients in whom positive fluorescence was detected out of the total number of patients. Only low-grade gliomas were considered, and data were grouped according to single fluorophores. Results: 16 papers about 5-aminolevulinic acid, 4 about sodium fluorescein, 2 about indocyanine green and 1 about tozuleristide were included in the systematic review. Regarding 5-aminolevulinic acid, a total of 467 low-grade glioma patients were included, and fluorescence positivity was detected in 34 out of 451 Grade II tumors (7.3%); while in Grade I tumors, fluorescence positivity was detected in 9 out of 16 cases. In 16 sodium fluorescein patients, seven positive fluorescent cases were detected. As far as indocyanine is concerned, two studies accounting for six patients (three positive) were included, while for tozuleristide, a single clinical trial with eight patients (two positive) was retrieved. Conclusions: The current evidence does not support the routine use of 5-aminolevulinic acid or sodium fluorescein with a standard operating microscope because of the low fluorescence rates. New molecules, including tozuleristide, and new techniques for fluorescence detection have shown promising results; however, their use still needs to be clinically validated on a large scale.

## 1. Introduction

Low-grade gliomas (LGGs) account for approximately 10–20% of all primary brain tumors [1]. Surgical resection is still the first-line treatment, and the extent of resection (EOR) has been shown to correlate positively with progression-free survival and quality of life [2,3,4,5,6,7]. A significant effort has been made to increase the EOR in glioma surgery; this has been facilitated by advances in intraoperative imaging, neuronavigation, optical visualization technology, and fluorescence-guided surgery (FGS) [8,9]. Over the past two decades, FGS has been increasingly used to support glioma surgery with the purpose of obtaining a maximal EOR [10,11]. This technique is considered a supplemental strategy to identify the intraoperatively of the tumor, as neoplastic cells have a higher fluorophore accumulation compared to normal brain tissue [12,13]. Currently, the most widely used fluorescent-labeling dye is 5-aminolevulinic acid (5-ALA), which is a naturally occurring metabolite of the heme synthesis pathway [10]. The strategy of using 5-ALA for FGS has demonstrated reassuring results in impacting progression-free survival and overall survival for patients with high-grade gliomas (HGG) [14,15,16,17,18,19,20,21]. FGS for LGGs [22] has a significantly smaller literature basis, and its role still needs to be established [23,24].

Despite these actual limits, the advancements in microscopic technology and the efforts to detect the presence of fluorophores at concentrations not perceptible by the surgeon’s vision may result in a possible beneficial incorporation of ALA FGS in the surgical management of LGGs [25,26].

A parallel current of research is exploring other interesting fluorescence agents that have a possible perspective application in this field [27]. For instance, sodium fluorescein (SF) has proved to be a marker for areas of compromised blood–brain barrier. This agent does not appear to selectively accumulate in glioma cells; rather, it locates in extracellular tumor cell-rich locations, thus representing a practical tool to help the surgeon identify the enhancing tumor regions [28]. Other agents have been proposed but not extensively assessed, such as indocyanine green (ICG), hypericin, 5-aminofluorescein-human serum albumin, and BLZ-100 (tozuleristide) [27,29].

An important research effort is required to establish the role of FGS in LGG management, considering the complete analysis of the current literature findings as the first essential step. The goal of this current review was to report the current knowledge about FGS in LGG and compare the available agents to understand which one would be more adaptable for surgical practice.

## 2. Materials and Methods

### 2.1. Search Strategy

A systematic review of the literature according to the Preferred Reporting Items 80 for Systematic Review and Meta-Analysis Protocols (PRISMA-P) guidelines was performed (Figure 1).

We searched two online databases—MEDLINE/PubMed and Embase—using the following free terms, keywords, or MeSH terms: “low grade glioma”, “fluorophore”, “fluorescence”, “aminolevulinic acid”, “indocyanine green”, “sodium fluorescein” with AND, OR, or NOT operators (Appendix A). All English language papers published over a 20-year period from January 2002 to December 2022 were considered. From the initial results page, we selected randomized and non-randomized studies and prospective and retrospective cohort studies. We excluded reviews, case reports, letters to the editor and editorials. Duplicates were removed. We did not request the missing study data from authors whose papers omitted such data. Titles and abstracts from the search results page were independently screened for eligibility by three review authors (A.B., P.Z. and M.B.) to identify the studies fulfilling the following inclusion criteria: adult population (>18 years) and papers considering LGG that report on the fluorescence rate (number of patients with positive fluorescence/total number of patients). Where both HGG and LGG were involved, the study was only included if the fluorescence rate was separated according to the histologic subtype. The definition of high-grade or low-grade glioma was determined based on the 2021 WHO classification of central nervous system (CNS) tumors [22] and the 2016 WHO CNS tumor classification for pre-2021 papers. Preclinical studies or studies not involving human subjects were excluded from the systematic review. Studies in which fluorophores were used with confocal microscopy and spectroscopy were not considered in the systematic review; however, their use was discussed in more detail in the discussion section. Due to the heterogeneity of fluorophores, we extrapolated and grouped data individually for each fluorophore (5-ALA, SF, ICG, BLZ-100). Each paper’s full text was read and critically evaluated by the same three review authors. Disagreements were resolved through consensus following a thorough discussion of the conflicting article. The reference lists were also screened to identify additional relevant papers.

### 2.2. Outcomes Definition

The main considered outcome was the fluorescence rate, defined as the number of patients in whom positive fluorescence was detected out of the total number of patients. Where available, the gross total resection (GTR) rate was reported. Considering the possible variability in the fluorophores uptake depending on the histologic subtype, where available, the different fluorescence rates were reported both according to the World Health Organization’s (WHO) grade and according to the specific histologic subtype itself. The risk of bias assessment is shown in Figure 2.

Considering the heterogeneity of the fluorophores addressed in this review, both the results and the discussion have been divided into sections according to the specific fluorescent agent: 5-ALA, SF, ICG or BLZ-100.

## 3. Results

The initial search resulted in a total of 927 papers. After excluding the duplicates, 44 papers published before 2002 and 9 non-English language papers, the titles and abstracts of 487 studies were evaluated. An additional 311 studies were excluded, and four reports were not retrieved, yielding 172 studies for the full-text evaluation. Of these 172 papers, 149 were excluded as they did not fulfill the inclusion criteria (Figure 1). Due to the lack of data about ICG, one paper by Haglund et al. was retrieved by cross-referencing and included in the systematic review, even though it was published in 1996. In the end, 16 papers about 5-ALA, 4 about SF, 2 about ICG and 1 about BLZ-100 were included in the systematic review. No studies about LGG were identified concerning other fluorophores like those of hypericin or 5-aminofluorescein-human serum albumin, which their use has only been studied in HGG, or endogenous fluorophores and new fluorescent molecules, which are still in preclinical phases. The characteristics of the selected studies are highlighted in Table 1.

### 3.1. 5-Aminolevulinic Acid

It is the most widely used fluorophore and whose use has been most investigated; in fact, 16 papers (six retrospective and nine prospective) were selected for the systematic review (Table 1). For quantitative analysis, only the studies in which the histologic findings were reported were considered, with a total of 467 LGG patients (451 WHO Grade II tumors and 16 Grade I tumors) (Table 2).

The mean fluorescence rate was 9.2%, being extremely variable among the case series (between 0% and 100%) because of the low number of patients in some of the case series considered and the histological heterogeneity of the patients themselves. Specifically, in Grade II tumors such as astrocytoma, oligodendroglioma, oligoastrocytoma and pleiomorphic xanthoastrocytoma, fluorescence positivity was found in 34 out of 451 cases (7.3%); while in Grade I tumors, such as dysembryoplastic neuroepithelial tumor (DNET), pilocytic astrocytoma, ganglioglioma and gemistocytic astrocytoma, fluorescence positivity was detected in 9 out of 16 cases (56.2%) (Figure 3).

### 3.2. Sodium Fluorescein

Four studies were included with a total of 16 patients, 13 Grade II tumors and 3 Grade I tumors. Concerning the Grade II tumors, 5 out of 13 patients (three astrocytomas and two oligodendrogliomas) were positive for fluorescence, whereas regarding the Grade I tumors, two out of two gangliogliomas had a positive fluorescence, while a non-uptake of SF by one pilocytic astrocytoma was reported (Table 2).

### 3.3. IndoCyanine Green

Two studies accounting for six patients were included. Among four Grade II astrocytomas, 50% showed fluorescence positivity, although it should be highlighted that in the two cases reported as positive, fluorophore administration had occurred prior to tumor excision, while in the two reported as negative, the drug was administered 24 h prior to surgery (ICG second window). Considering Grade I tumors with the indocyanine second window, one pilocytic astrocytoma did not uptake ICG, whereas fluorescence was evidenced in one ganglioglioma.

### 3.4. BLZ-100 (Tozuleristide)

As far as tozuleristide is concerned, a single clinical trial with eight patients, including six oligodendrogliomas, one oligoastrocytoma (Grade II) and one pilocytic astrocytoma, was considered. Among the WHO Grade 2 tumors, fluorescence positivity was shown in two cases (both oligodendrogliomas) out of seven patients, corresponding to 28.5%. In contrast, no positivity was shown for a pilocytic astrocytoma.

## 4. Discussion

Fluorescence-guided surgery (FGS) has developed over the last two decades as a formidable augmentation strategy to promote maximal safe resection in glioma surgery. Most of the literature supporting this modality concerns the setting of high-grade gliomas. Indeed, the role of fluorescence-guided surgery in low-grade gliomas has not been established yet [13].

### 4.1. 5-Aminolevulinic acid

Based on the consolidated results obtained by 5-ALA FGS in HGG, its application also gained increasing interest for LGG surgery [51]. Anyway, despite the high volume of literature concerning 5-ALA for FGS in HGGs, the extent of investigation for LGGs remains sparse within the literature [52].

The first study to assess a clinical application of ALA in LGG was Widhalm et al., who published results that included eight patients with WHO Grade II gliomas, in which none of the patients demonstrated fluorescent foci [30]. Similar results were observed in a study by Tsugu et al., which included six patients with WHO Grade II gliomas, none of which demonstrated ALA-included fluorescence intraoperatively [31]. Subsequent results obtained by the application of 5-ala FGS in LGG surgery have demonstrated a positive fluorescence rate within 10–20% of cases while utilizing a standard operating microscope [11,12,13,14,15,16,17,18,19].

In a study conducted by Ewelt et al., only one of eleven WHO Grade II gliomas showed visible fluorescence with 5-ALA administered preoperatively [33]. Further evaluation of ALA-induced fluorescence was undertaken by Saito et al., including 42 patients in the analysis, eight of which had WHO Grade II gliomas [34]. Under standard operating conditions, two of eight patients with WHO Grade II gliomas demonstrated fluorescence intraoperatively [34]. Widhalm et al. further investigated the utility of ALA FGS through a larger patient population than the 2010 study, including 59 patients with non-contrast enhancing lesions, 33 of which were WHO Grade II gliomas [32]. The result of this analysis was similar to the previous one: a minority of lesions classified as a WHO Grade II diagnosis revealed fluorescent foci (4 of 33) [32].

Considering bigger patient cohorts, Jaber et al. evaluated a 166-patient population, 82 of whom were diagnosed with WHO Grade II gliomas [24]. Fluorescence was classified into four categories according to the pattern observed intraoperatively: absent, patchy-weak, homogenous-weak or strong. For LGGs, 83.1% demonstrated no fluorescence, 6.1% demonstrated patchy-weak fluorescence, 4.9% weak-homogenous, and 4.9% demonstrated strong fluorescence under standard operating conditions [24]. Considering all the lesions with any positive fluorescence, about 16% of the tumors were detectable using FGS [24]. In a subsequent analysis from the same group of a cohort of 74 patients with histologically verified LGG, 16 (22%) patients revealed visible intraoperative fluorescence [37].

A recent study by Hosmann et al. [35], with the use of 5-ALA in LGG, showed that visible fluorescence was detected in 7 (12%) of the 59 included patients in focal intratumoral areas. The result was further confirmed by the same authors in 2022 with a retrospective study that included 86 LGGs [36]. In this scenario, visible fluorescence was found during surgery in 13 (15%) cases [36]. An investigation of 5-ALA FGS by Marbacher et al. included 531 total cases of intracranial tumors that underwent resection or biopsy [38]. The LGG population in this analysis included 17 WHO Grade II and 3 WHO Grade I gliomas, which demonstrated ALA-positive foci in 40% under standard operating microscope conditions. All the positive cases in this group were from WHO Grade II glioma tissue [38]. The higher rate of positive fluorescence rates may be related to the choice of considering every positive lesion with even minimal fluorescence, in addition to intratumor heterogeneity and sampling bias [38]. In a further study, Goryaynov et al. reported the presence of visible 5-ALA fluorescence in a markedly higher portion of LGG patients (52%) [50]. However, this study not only included diffusely infiltrating WHO Grade II gliomas but also pilocytic astrocytomas, gemistocytic astrocytomas and desmoplastic infantile gangliogliomas [50]. When analyzing different subgroups separately, only 29% of WHO Grade II astrocytomas showed visible fluorescence, which is in accordance with earlier publications [51]. Kaneko et al. observed visible fluorescence in 39% of the evaluated patients with LGG, which is higher than in previous reports [39]. Anyway, the authors suggest that this rate may be related to their case selection: they mainly included larger tumors, tumors with a positive 18F-FET signal and tumors with enhancement [39].

Smaller series may report higher fluorescence rates; however, these results are influenced by patient selection, as in the study conducted by Schebesch et al. and the study by Chan et al., both showing a fluorescence rate of 100% for LGGs [40,41]. Some studies obtained better results in terms of the fluorescence rate; however, in many cases, it is possible to explain these data by analyzing the patients’ selection criteria [38,39,50].

Surprisingly, no study to date inspecting ALA-derived PpIX fluorescence in LGG has investigated higher doses than the labeled dose of 20 mg/kg. Also, the investigators adhered mostly to a specific 3- to 4-h time frame of administering the drug preoperatively [53]. A recent study by Kaneko et al. quantified the visible fluorescence intensity over time and analyzed the time-based dynamic of PpIX in LGG tissue [39]. Time kinetics of fluorescence in this ex vivo study revealed that the highest fluorescence intensity and PpIX concentration were observed 7–8 h after 5-ALA administration [39]. This result is in accordance with the results obtained for HGG, suggesting that tumor cells in LGG follow the same kinetics as in HGG. The authors suggest that optimizing surgery according to the time dependency of this fluorophore might improve fluorescence visualization, especially in weakly fluorescing tumors [39]. Further studies that investigate different 5-ALA doses and timings of administration in LGG are needed [53].

#### Histology-Related Considerations

Gliomas are known to demonstrate great heterogeneity within the same tumoral lesion [54,55]. The observable heterogeneous distribution of PpIX fluorescent foci within LGG is hypothesized to represent neoplastic regions with higher rates of cell proliferation and a greater propensity for malignant transformation [19,30,56]. More specifically, there is a significant correlation between visible 5-ALA fluorescence with histopathological criteria of malignancy, such as the mitotic rate, cell density, nuclear pleomorphism and proliferation rate [32,57]. Given these findings, it was hypothesized that visible 5-ALA fluorescence in LGGs might be an indicator of aggressive tumor behavior [58].

The correlation between increased cell density and fluorescence intensity was originally reported by Stummer et al. [56]. In the study from Widhalm et al., the lesions with negative ALA fluorescence demonstrated lower MIB-1 labeling and lower PETmax, compared to the more malignant WHO Grade III gliomas, supporting that, with lower features of malignancy, there is less frequent ALA fluorescence labeling [30]. These initial findings, suggesting that visible 5-ALA fluorescence in suspected LGG is a predictive marker for high-grade histology, were confirmed in an independent cohort by the study conducted by Ewelt et al. [33]. Of the 30 included tumors, 13 were diagnosed as WHO Grade II, 15 as WHO Grade III and 2 as WHO Grade IV gliomas [33]. The majority of WHO Grade III and IV gliomas demonstrated visible fluorescence (2 of 2 GBM, 10 of 15 WHO Grade III gliomas; 70%), while only 1 of 11 WHO Grade II gliomas showed visible fluorescence [33].

In the study of Jaber et al. [37], patients were analyzed for 5-ALA fluorescence related to the outcome. In the case of positive fluorescence, patients had a shorter time to malignant transformation and a shorter overall survival (over a median follow-up period of 48 months) [37]. Also, in the study by Hosmann et al., patients with fluorescing lesions had significantly shorter PFS, MTFS and OS compared to patients with non-fluorescing tumors [35].

In 2013, Widhalm et al. performed a histological evaluation of ALA-fluorescent foci within WHO Grade II and III lesions, demonstrating that the fluorescent regions correlated with the mitotic rate, cell density, nuclear pleomorphism and MIB-1 labeling index [32]. Additionally, this study reported a significant correlation of focal 5-ALA fluorescence with specific histopathological parameters of anaplasia [32]. In 2019, Martinez-Moreno et al. found a significant correlation between the fluorescence intensity values and histological parameters of malignancy and proliferation using ex vivo spectroscopic analysis [59]. A following study from Jaber et al. reported a higher EGFR expression in patients with fluorescing tumors (*p*  =  0.056), suggesting that visible fluorescence might be an indicator for an angiogenic switch and incipient malignant transformation [37]. Recently, Hosmann et al. studied the correlation between 5-ALA fluorescence and neoangiogenesis by investigating the expression of CD34 in fluorescent areas [36]. CD34 is considered an established marker for vascular endothelial progenitors, potentially indicating tumor progression. These data indicate that CD34 expression is associated both with intraoperative 5-ALA fluorescence and outcomes in patients with LGG [36]. Thus, visible fluorescence might be an intraoperatively available marker of unfavorable patient outcomes, indicating tumor areas that are more aggressive and characterized by vascular proliferation [35,36]. Only one study reported that the Ki-67/MIB-1 index is not significantly different between the WHO Grade II fluorescent and non-fluorescent groups [24]. This result was obtained by Jaber et al., considering 82 LGGs [24]. Surprisingly, in the quantitative analysis we conducted, a higher rate of fluorescence is shown in Grade I gliomas, although no conclusions can be drawn since the number of patients is small. In any case, this finding suggests that in addition to the malignancy characteristics, the fluorescence rate is also related to tumor-specific factors.

The results of these studies further support the role of ALA FGS in preventing the risk of under grading gliomas: the surgeon can use fluorescence to accomplish an accurate sampling of anaplastic foci. Moreover, this might also serve to identify patients that should be considered for aggressive adjuvant therapy in the immediate postoperative period [13].

### 4.2. Sodium Fluorescein

Different to 5-ALA, sodium fluorescein does not selectively accumulate in glioma cells, showing a low specificity rate. It indeed shows a sensibility of 100% for the real-time identification of regions of blood–brain barrier breakdown [60], which is not usually found in LGG. For these reasons, there are few studies reporting results about the use of fluoresceine in LGG, mainly with a low number of patients included.

The great sensibility of fluorescein for gliomas is confirmed in a study by Nevzati et al., in which the authors assessed the reliability of fluorescein sodium in guiding stereotactic biopsies. All tumors could be detected with fluorescein fluorescence in that series, thereby leading to a sensitivity of 100% and a positive predictive value of 94% [42]. Anyway, the number of patients with LGG in this study was low (three cases), and the selection criteria included contrast enhancement in MRIs, which is not a common trait in LGGs.

Schebesch et al. [40] reported a series of five patients that received a low dose of sodium fluorescein (5 mg/kg); two of them were confirmed as LGG. All the cases demonstrated some degree of fluorescence, which correlated with PET metabolic activity but not with MRI contrast enhancement. One Grade II tumor was visible only under the fluorescence-detecting filter. Nonetheless, Schebesch et al. [40] concluded that the incorporation of SF was helpful both in detecting lesions and in ascertaining their borders, regardless of their grade. In 2018, Xiang et al. performed FGS with low-dose SF (5 mg/kg) in a series of 28 patients with suspected glioma. Of these 28 patients, there were five LGGs, but none of them revealed positive yellow fluorescence intraoperatively [43]. The authors conclude that fluorescein-guided resection is an effective and convenient technique for HGG surgery but not for LGG surgery [43].

A randomized control trial was conducted by Chen et al. to compare the results achieved in glioma surgery with and without the use of fluorescein [44]. The study group was composed of 10 patients (4 patients were confirmed LGGs) that received intravenous injections of high-dose SF (15–20 mg/kg); the control group consisted of 12 patients that did not receive fluorescent agents during surgical resection [44]. Significant differences were observed between the two patient groups in the gross total resection (GTR) rates (Fisher’s exact test *p* = 0.047) and progression-free survival (Student’s *t*-test *p* = 0.033), with better results obtained for the study group [44].

Due to the different mechanisms of action, which make it less ideal for LGGs, few studies have been conducted, and with fewer patients than 5-ALA, poor conclusions can be drawn. However, the results may depend on the dose of fluoresceine and the additional use of filters or confocal microscopy, and new studies should be guided in these directions.

### 4.3. IndoCyanine Green

Indocyanine Green (ICG) is one of several near-infra-red (NIR) fluorescent agents; however, it is the only NIR-fluorescent contrast agent approved for use in humans [61]. It is low in cost, minimally toxic, and allows real-time visualization of tumoral lesions. Several groups have experimented with ICG with an intravenous injection of the agent before resection [62,63,64,65]. ICG has a peak excitation of 780–790 nm and an emission spectrum of 805–820 nm in human tissues.

At first, Haglund et al. [45] studied the possible use of ICG in human glioma in 1996. They included nine patients, and two of them were confirmed LGGs [45]. The authors noted differences in the uptake and clearance of the dye, even if the time to observe the events was very short [45].

Ferroli et al. [66] evaluated the possible role of ICG video angiography in providing information about the vascular physiopathology of CNS tumors. They studied a heterogeneous population, including also 17 LGGs. ICG video angiography allowed for the intraoperative real-time assessment of exposed vessels with excellent image quality and resolution; however, it was unable to define the margins of the tumor.

As noted in previous studies, ICG provides vascular enhancement at the surface of the tumor; however, the dye rapidly clears and leaves significant background noise [46]. Lee et al. [46] have been studying alternative approaches using NIR dyes to develop a contrast between tumor tissue and the surrounding normal brain parenchyma, finally developing the “Second Window ICG” method. In this case, ICG is solubilized in a higher concentration using sodium chloride and administered 24 h prior to imaging. Over 24 h, the dye accumulates in the tumor tissue because of the enhanced permeability and retention (EPR) effect (small molecules may pass through a disrupted BBB and be retained due to a relative lack of drainage) and thanks to a relatively hypoxic microenvironment [67,68]. Lee et al. conducted a study on 12 patients (4 patients had a confirmed LGG, 2 had a WHO Grade I, and 2 had a WHO Grade II) to analyze the results obtained using their technique [46]. The authors report strong tumor-to-background fluorescence ratios and little auto-fluorescence or background noise while identifying tumor margins. Thus, the fluorescent signal was visible prior to opening the dura (61%), became stronger at the cortical surface after opening the dura (77%), and demonstrated a strong (100%) signal when finally exposing the tumor via corticectomy. Anyway, only two of the four LGGs were visible with NIR and only at the last step (tumor exposure). Therefore, the authors state that Second Window ICG may provide a practical and sensitive means of identifying glioma at different surgical steps. Its use for margin detection and establishing EOR remains under study [46].

### 4.4. BLZ-100 (Tozuleristide)

Tozuleristide is a near-infrared (NIR) tumor-targeting molecule composed of a combination of chlorotoxin (CTX) and ICG. CTX is a peptide that demonstrated excellent binding to multiple tumor types, including high- and low-grade gliomas, and it was tested in human phase I trials with no appreciable toxicity or side effects [69]. The extent of CTX binding to glioma cells, as measured by immunohistochemistry, has been shown to correlate with the histological grade: the binding rate is about 40–45% in LGGs [70].

Therefore, additional studies are required to test its possible role for FGS in gliomas in conjunction with a novel, highly sensitive NIR imaging device (Synchronized Infrared Imaging System, SIRIS) [69]. Patil et al. conducted a phase I trial that demonstrated a reliable uptake and the ability to differentiate tumors from normal brain tissue [47]. They included 17 adult subjects with glioma, with nine HGG and eight LGG cases. At doses of 9 mg and above, clear evidence of tozuleristide uptake into tumors was observed in 9 of the 14 cases (64%), with a 25% fluorescence positivity in LGGs [47]. They proved that the method had good accuracy, with an absence of fluorescence in normal brain tissue and a correlation of fluorescence with the presence of a tumor (confirmed in the histological study) [47]. Currently, a phase II study is ongoing; however, it is performed on pediatric populations affected by glioma (NCT03579602).

Even if we cannot draw conclusions on the possible use of BLZ-100 in glioma surgery, these encouraging results justify further investigations and future trials using this fluorescence agent.

### 4.5. Confocal Microscopy

In the reported studies about 5-ALA-guided surgery, diagnostically significant PpIX fluorescence is achieved in about 10–20% of LGGs [11,12,13,14,15,16,17,18,19]. It is hypnotized that fluorescence may not be absent in these lesions; however, it cannot be observed because its intensity is below the detection threshold for standard operating microscopes. This highlights the importance of augmentation techniques to empower PpIX fluorescence detection abilities in the setting of LGGs [13,71].

Therefore, intraoperative confocal microscopy has been used for the detection of PpIX fluorescence in LGG surgery [26]. Sanai et al. reported a 100% positive fluorescence rate in their series of 10 LGG cases visualized intraoperatively under confocal microscopy [26]. The confocal microscopic analysis for the presence of fluorescence was performed upon initial tumor confrontation, at the midpoint of resection, and at the presumed brain–tumor interface for 10 patients with WHO Grade I (n  =  1) and II (n  =  9) lesions. Confocal microscopy for FGS provided an accurate method of visualizing fluorescence in neoplastic cells within the glioma and, importantly, along the brain–tumor interface [26]. Additional studies may be useful to further evaluate this tool in larger patient cohorts. Additionally, to empower the support provided by confocal microscopy probes, more sophisticated instruments have been developed and continue to be refined for high-speed visualization of PpIX fluorescence in gliomas [71].

Martirosyan et al. tried to evaluate the possible role of confocal laser endomicroscopy (CLE) in glioma FGS after the administration of low-dose SF (5 mg/kg) [72]. In 66 patients, they identified eight Grade II gliomas. They concluded that SF contrast with CLE is inadequate for defining histologic characteristics and the grade definition in glioma; however, it provides a real-time in vivo identification of tumor areas, substantially improving intraoperative decisions [72]. Similarly, Pavlov et al. conducted a study using low-dose SF and confocal microscopy. Their aim was to verify the feasibility of intraoperative in vivo probe-based confocal laser endomicroscopy (pCLE) in the surgery and biopsy of gliomas with the use of fluorescent agents. In their series, including two LGGs, fluorescein allowed a good differentiation between normal and pathological tissue. This differentiation was true whether the procedure was a stereotactic biopsy or open surgery [73].

A case report from Belykh et al. further explored the use of CLE with high-dose SF (40 mg/kg) for LGG. The authors support the choice of high-dose SF instead of low-dose, claiming that low-dose SF caused suboptimal, uninterpretable, or too-dark CLE imaging in their previous experience [74]. This report suggests that, given the highly informative images, the minimal adverse effects of SF, and the good postoperative outcome of this case, higher doses of SF (20–40 mg/kg) should be considered or trialed with intraoperative CLE [74].

### 4.6. Laser Spectroscopy

In recent years, novel technologies for improved fluorescence detection in brain tumors have been introduced into the neurosurgical field [75]. One of the most promising methods represents a quantitative spectroscopic analysis of 5-ALA-induced PpIX accumulation. Using this technique, a hand-held, fiber-optic probe connected to a spectrometer measures the PpIX accumulation and fluorescence in real time. In this sense, spectroscopic analysis is capable of measuring the characteristic fluorescence signal of PpIX with typical emission peaks at 635 nm and 710 nm [76,77,78]. The first study was performed by Utsuki et al. in 2006, analyzing the potential of intraoperative laser spectroscopy in six patients with brain tumors [78]. This promising approach was able to detect tumor tissue by laser spectroscopy at the peak of 636 nm after 5-ALA administration, despite the absence of visible fluorescence [78].

It was then proposed to use this instrument in LGG tumors that do not show visible fluorescence by using available optical technologies [79]. Ishihara et al. reported an ex vivo quantitative analysis of 5-ALA-induced PpIX fluorescence intensity using spectroscopy in 2007 [76]. In this study, 65 samples of six glioma patients (two that were WHO Grade II, two were WHO Grade III and two were WHO Grade IV gliomas, respectively) were quantitatively analyzed and correlated with the fluorescence intensity and specific histopathological criteria [76]. They were the first to report that the tumor tissue of LGGs with the absence of visible fluorescence could be detected by quantitative spectroscopic analysis of fluorescence intensity [76].

FGS with 5-ALA was analyzed in 2011 by Valdes et al. for multiple intracranial tumor types, including two patients with LGG [77]. They used an intraoperative fiberoptic probe connected to a spectrometer to quantitate the fluorescence intensity following stimulation of the tissue with blue light [77]. The findings from this study demonstrated a statistically significant difference in the concentration of PpIX between tumor samples and native brain parenchyma, which was also confirmed for LGG. Therefore, this first in vivo study demonstrated the feasibility of this innovative approach [77]. However, this study had an extremely small population size (n  =  2 for LGG); therefore, selection bias is an important limit in this case. A following study from the same group showed a positive fluorescence rate of 42% in a group of 12 LGGs using the fiberoptic probe [48]. The authors suggest that low levels of PpIX commonly accumulate in LGGs: the marker is diagnostically significant, even if below the detection threshold of current visual fluorescence techniques [48].

Based on these promising preliminary data, Widhalm et al. applied this fiberoptic probe in addition to conventional visual 5-ALA fluorescence technology in a study published in 2019 during the surgeries of 22 suspected diffusely infiltrating LGGs [49]. While visible fluorescence was absent in all LGGs, the detection rate raised to 75% with the use of the fiberoptic probe [49]. The authors of this study thus conclude that the additional use of quantitative PpIX analysis by the fiberoptic probe represents a powerful technique: it allows improvement in the intraoperative detection of LGG tissue that is generally characterized by the absence of visible fluorescence [49].

## 5. Conclusions

Current evidence, particularly regarding 5-ALA and SF, does not support the routine use of these fluorophores with a standard operating microscope because of the low fluorescence rates. However, they are still a possible candidate for future improvements in LGG surgery results, and further studies should explore new strategies to obtain higher fluorescence rates and differentiate the fluorescence on the basis of histological type. In particular, the possibility that 5-ALA-capturing tumors have a higher level of aggressiveness, which may be underestimated by histologic grading, may be explored. Such studies should focus on more sensitive instruments for porphyrin detection, as shown by the encouraging results of confocal microscopy and laser spectroscopy and alternative dosage regimes for both 5-ALA and SF administration. New molecules, including tozuleristide, have shown promising results; however, their use still needs to be clinically validated on a large scale, particularly for LGGs.

## Figures and Tables

**Figure 1 cancers-15-04130-f001:**
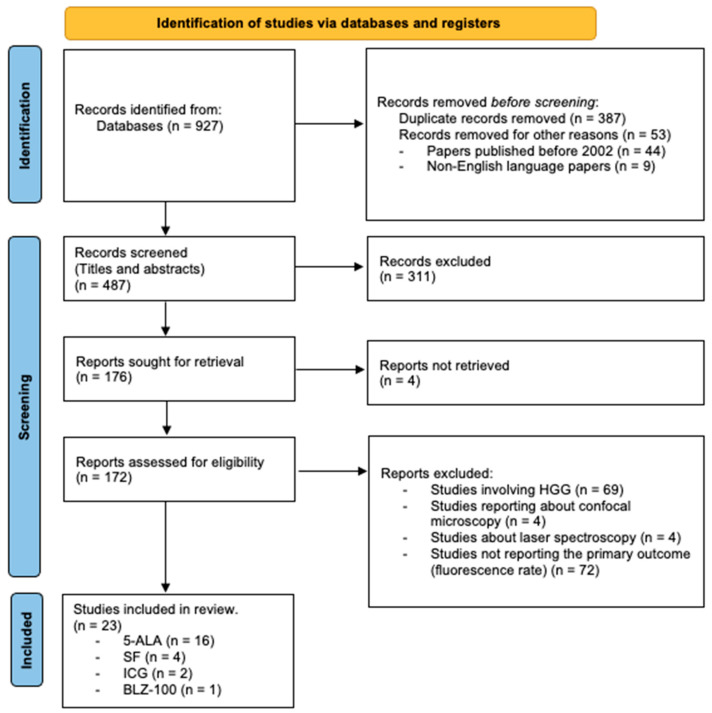
PRISMA flow chart.

**Figure 2 cancers-15-04130-f002:**
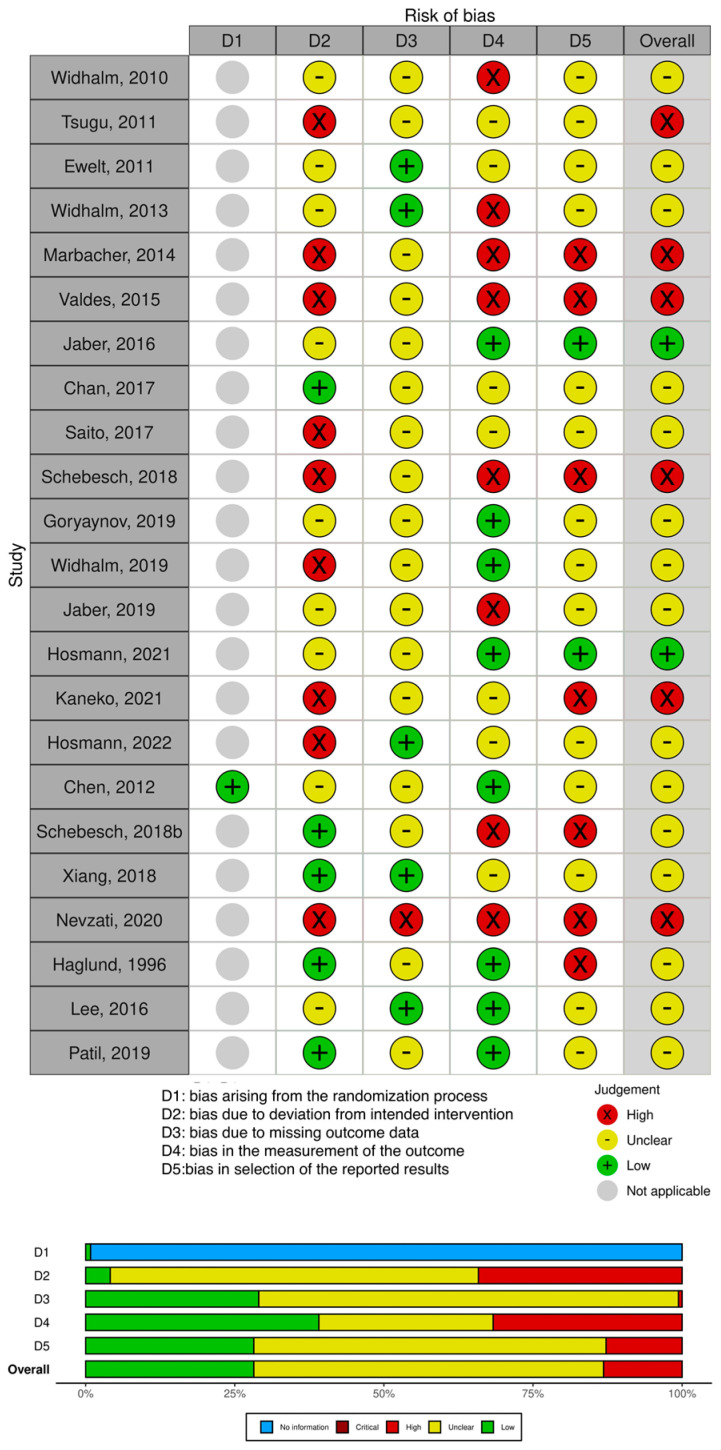
Risk of bias assessment for the included studies [24,30,31,32,33,34,35,36,37,38,39,40,41,42,43,44,45,46,47,48,49].

**Figure 3 cancers-15-04130-f003:**
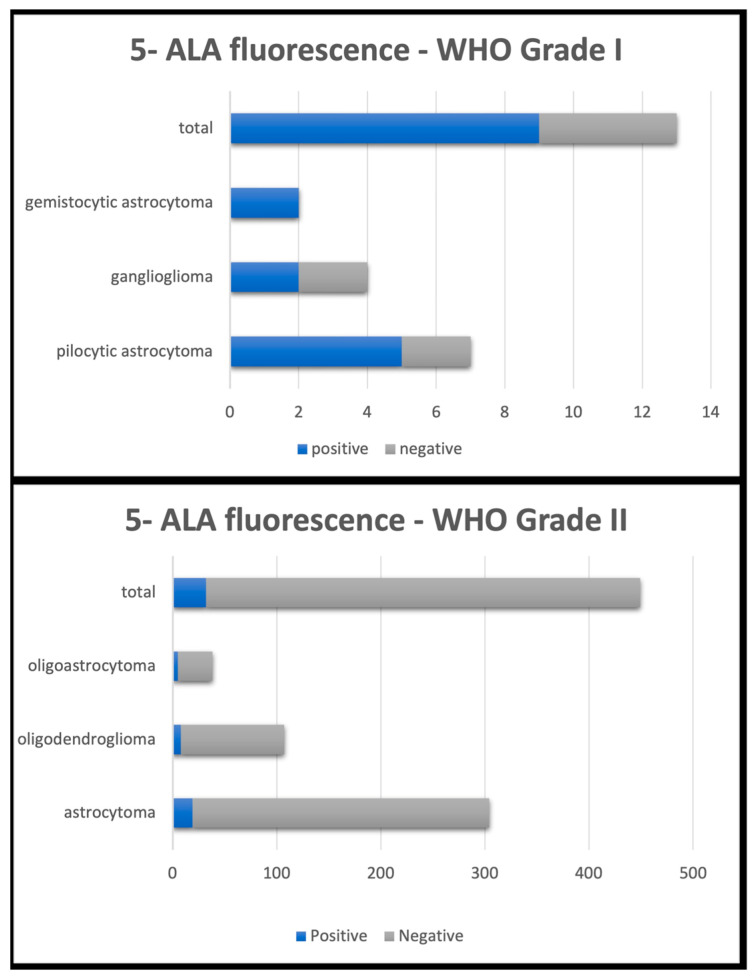
5-aminolevulinic acid fluorescence rates according to WHO grade and histologic subtype.

**Table 1 cancers-15-04130-t001:** Characteristics of the selected studies.

Author, Year	Type of Study	n. of Patients	Fluorophore	Control Group	Fluorescence Rate	GTR Rate	Notes	Histology
Widhalm, 2010 [30]	prospective	8	5-ALA	no	0%	82%		yes
Tsugu, 2011 [31]	retrospective	6	5-ALA	no	0%	47%	with intraop MRI	no
Ewelt, [33] 2011	prospective	13	5-ALA	no	8%	100%		yes
Widhalm, 2013 [32]	prospective	33	5-ALA	no	12%	64%		yes
Marbacher, 2014 [38]	retrospective	20	5-ALA	no	40%	not reported	12/17 biopsies	yes
Valdes, 2015 [48]	prospective	12	5-ALA	no	42%	not reported		yes
Jaber, 2016 [24]	prospective	82	5-ALA	no	16%	not reported		yes
Chan, 2017 [41]	retrospective	3	5-ALA	no	100%	56%		no
Saito, 2017 [34]	prospective	8	5-ALA	no	25%	not reported		yes
Schebesch, 2018 [40]	prospective	2	5-ALA	no	100%	not reported		yes
Goryaynov, 2019 [50]	prospective	27	5-ALA	no	52%	not reported		yes
Widhalm, 2019 [49]	prospective	8	5-ALA	no	0%	not reported		yes
Jaber, 2019 [37]	retrospective	74	5-ALA	no	22%	32%		yes
Hosmann, 2021 [35]	retrospective	55	5-ALA	no	12%	49%		yes
Kaneko, 2021 [39]	retrospective	21	5-ALA	no	36%	n.a.	biopsies	yes
Hosmann, 2022 [36]	retrospective	86	5-ALA	no	15%	57%		yes
Chen, 2012 [44]	randomized controlled trial	5	SF	yes	75%	80% (33% in control group)	High dose (15–20 mg/kg)	yes
Schebesch, 2018 [40]	case series	2	SF	no	100%	not reported		yes
Xiang, 2018 [43]	prospective	5	SF	no	0%	74%		yes
Nevzati, 2020 [42]	retrospective	4	SF	no	0%	n.a.	stereotactic biopsies	yes
Haglund, 1996 [45]	case control	2	ICG	yes	100%	not reported	ICG before resection	yes
Lee, 2016 [46]	prospective	4	ICG	no	50%	25%	ICG second window	yes
Patil, 2019 [47]	prospective	8	BLZ-100	no	25	not reported		yes

**Table 2 cancers-15-04130-t002:** Fluorescence rates divided by fluorophores and histologic subtype.

		5-ALA			SF			ICG			BLZ-100		
		**yes**	**no**	**tot**	**yes**	**no**	**tot**	**yes**	**no**	**tot**	**yes**	**no**	**tot**
*Grade II*	astrocytoma	19	285	304	3	5	8	2	2	4	na	na	
	oligodendroglioma	8	99	107	2	3	5	na	na		2	4	6
	oligoastrocytoma	5	33	38	na	na		na	na		0	1	1
	pleomorphic xanthoastrocytoma	2	0	2	na	na		na	na		na	na	
	**total, %**	**34 (7.5%)**	**417**	**451**	**5 (38.4%)**	**8**	**13**	**2 (50%)**	**2**	**4**	**2 (28.5%)**	**5**	**7**
*Grade I*	pilocytic astrocytoma	5	2	7	0	1	1	0	1	1	0	1	1
	ganglioglioma	2	2	4	2	0	2	1	0	1	na	na	
	gemistocytic astrocytoma	2	0	2	na	na		na	na		na	na	
	DNET	0	3	3	na	na		na	na		na	na	
	**total, %**	**9 (56.2%)**	**7**	**16**	**2 (66.7%)**	**1**	**3**	**1 (50%)**	**1**	**2**	**0 (0%)**	**1**	**1**

## Data Availability

No new data were created or analyzed in this study. Data sharing is not applicable to this article.

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
