# Peer review of "How Reliable Is Fluorescence-Guided Surgery in Low-Grade Gliomas? A Systematic Review Concerning Different Fluorophores"

_cancers, 2023, doi:10.3390/cancers15164130_

Round 1

Reviewer 1 Report

Bianconi et al describe a review for fluorescence-guided surgery with some fluorophores in low grade glioma at 2002-2022. The review is well written and seems to have few issues. Search methods are also probably appropriate. After minor revision, the manuscript may be suitable for publication in the journal.

1.         Avoid abbreviations in Abstract, e.g., PRISMA and LGG.

2.         Line 80: “PRISMA-P (Preferred Reporting Items 80 for Systematic Review and Meta-Analysis Protocols) guidelines” is replaced by “Preferred Reporting Items 80 for Systematic Review and Meta-Analysis Protocols (PRISMA-P) guidelines”.

3.         Line 112: World Health Organization (WHO).

4.         Please replace by Fig 2 with high resolution. The color scheme of the bottom panel is unknown.

5.         It should be explained that diagnosis is based on the diagnostic criteria in the literature. Besides, it may be appropriate to explain the latest WHO classification 2021 (ref 22).

Author Response

We thank the reviewer for the observations and comments. 

  1. Avoid abbreviations in Abstract, e.g., PRISMA and LGG.

The abbreviations were deleted.

  1. Line 80: “PRISMA-P (Preferred Reporting Items 80 for Systematic Review and Meta-Analysis Protocols) guidelines” is replaced by “Preferred Reporting Items 80 for Systematic Review and Meta-Analysis Protocols (PRISMA-P) guidelines”.

The requested change was made.

  1. Line 112: World Health Organization (WHO).

The requested change was made.

  1. Please replace by Fig 2 with high resolution. The color scheme of the bottom panel is unknown.

The quality of figure 2 has been reduced in order to be included in the word file. We have uploaded the original full resolution version in the supplementary files

  1. It should be explained that diagnosis is based on the diagnostic criteria in the literature. Besides, it may be appropriate to explain the latest WHO classification 2021 (ref 22).

We added the diagnostic criteria we used in the systematic review in Materials and Methods section

“The definition of high-grade or low-grade glioma was determined based on the 2021 WHO classification for central nervous system (CNS) tumors [22], and WHO 2016 CNS tumor classification for pre-2021 papers”

Reviewer 2 Report

The review serves as a good summary of fluorophores used in low grade gliomas surgery. The conclusion is well supported by the reviewed cases and authors' analyses, that fluorescence-guided surgery in low grade gliomas is not reliable due to the low fluorescence rates but witness on-going development. A few questions:

1. in the case of 5-Aminolevulinic acid, authors mentioned cases with good fluorescence rates and postulated this is due to case selection (line 230-239). Does that mean in certain cases, 5-ALA is somehow reliable? If so, could we figure out under which circumstances with available data?

2. The histological application of 5-ALA in terms of identifying undergrading gliomas is intriguing. Should we mentioned it in the conclusions?

Author Response

We thank the reviewers for the comments and observations.

The review serves as a good summary of fluorophores used in low grade gliomas surgery. The conclusion is well supported by the reviewed cases and authors' analyses, that fluorescence-guided surgery in low grade gliomas is not reliable due to the low fluorescence rates but witness on-going development. A few questions:

  1. in the case of 5-Aminolevulinic acid, authors mentioned cases with good fluorescence rates and postulated this is due to case selection (line 230-239). Does that mean in certain cases, 5-ALA is somehow reliable? If so, could we figure out under which circumstances with available data?

I don't think we can say that 5-ALA in some cases is reliable, but rather that in some cases (eg. tumors that take contrast on MRI although on histologic examination they are considered as low grade, with uptake at 18F Dopa PET, with a higher Ki67 index) it may show a higher fluorescence rate. That is to say that on the basis of the data from this review, 5-ALA cannot be used with confidence to define tumor margins, unless it is used with fluorescence enhancement techniques, as much as to give a clue to the greater aggressiveness of the uptaking tumor.

  1. The histological application of 5-ALA in terms of identifying undergrading gliomas is intriguing. Should we mentioned it in the conclusions?

We agree with the reviewer and have added this sentence in the conclusion

In particular, the possibility that 5-ALA capturing tumors have a higher level of aggressiveness that may be underestimated by histologic grading may be explored.